# Research on Improved DenseNets Pig Cough Sound Recognition Model Based on SENets

**Hang Song, Bin Zhao \*, Jun Hu, Haonan Sun and Zheng Zhou**

College of Engineering, Heilongjiang Bayi Agricultural University, Daqing 163319, China
* Correspondence: zhaobin@byau.edu.cn; Tel.: +86-13634598552

**Abstract:** In order to real-time monitor the health status of pigs in the process of breeding and to achieve the purpose of early warning of swine respiratory diseases, the SE-DenseNet-121 recognition model was established to recognize pig cough sounds. The 13-dimensional MFCC, ΔMFCC and $\Delta^2$MFCC were transverse spliced to obtain six groups of parameters that could reflect the static, dynamic and mixed characteristics of pig sound signals respectively, and the DenseNet-121 recognition model was used to compare the performance of the six sets of parameters to obtain the optimal set of parameters. The DenseNet-121 recognition model was improved by using the SENets attention module to enhance the recognition model's ability to extract effective features from the pig sound signals. The results showed that the optimal set of parameters was the 26-dimensional MFCC + ΔMFCC, and the rate of recognition accuracy, recall, precision and *F1 score* of the SE-DenseNet-121 recognition model for pig cough sounds were 93.8%, 98.6%, 97% and 97.8%, respectively. The above results can be used to develop a pig cough sound recognition system for early warning of pig respiratory diseases.

**Keywords:** porcine respiratory disease; porcine cough sound recognition; DenseNet; SENets





## 1. Introduction

With increased market demand, pork has become the most consumed meat in the world, and the impact of pork as a sustainable livestock product is critical to global food security [1,2], but the expansion of farming makes pigs susceptible to respiratory diseases. Respiratory diseases in pigs can reduce the immunity of the affected pigs, which in turn leads to death and reduced productivity, affecting the economic efficiency of the pig farming industry [3–7]. Early warning of respiratory diseases in pigs can improve the above problems. The sound signal of the pig cough sound can be used as the main basis for screening and diagnosis of early respiratory diseases in pigs [8–10]. Therefore, the key to achieving intelligent early warning of respiratory diseases in housing pigs is to accurately identify the pig cough sound.

At first, some scholars studied the acoustic features of pig sound. Sara [10,11] found in her study that the Root Mean Square (RMS), peak frequency (Hz) and cough interval of healthy pigs were only significantly different from those of diseased pigs, indicating that acoustic parameters would change with the health status of pigs. In a study of the dynamics in the energy envelope of pig cough sounds, Mitchell et al. [12,13] found that there were significant differences in the dynamic changes of the short-term energy envelope between the produced induced coughs via nebulization of citric acid and the pathological pig cough sound, which indicates that the information characteristics of pig sound energy can be used to reflect the respiratory health of pigs. The findings of these studies laid the foundation for the subsequent research of a pig cough sound recognition algorithm.

Subsequently, more scholars have made certain achievements in pig sound recognition. Hirtum et al. [14] induced a physiological cough in six piglets via citric acid atomization, applied distance function to fast Fourier spectral sound analysis, and studied the dichotomous classification of "cough" and "other". The recognition accuracy of the pig cough sound

is 92%, but there is still 21% misclassification for the whole sound database. Exadaktylos et al. [15] used a fuzzy c-means algorithm to classify pig sound signal samples. The square Euclidean distance was used to determine the threshold, resulting in an 82% recognition rate of diseased pigs and an 85% overall recognition rate. Alexandra et al. [16] established a decision tree algorithm based on machine-learning [17,18] technology according to the differences in different frequency characteristics of pig sounds to judge the emotions of pigs. The recognition accuracy of pig distress was 81.92%.

In recent years, deep learning algorithms have been widely used in the field of artificial intelligence [19–22]. Li et al. [23] used them in the study of pig cough sound recognition; they used Deep Belief Nets (DBN) and constructed a 5-layer pig cough sound recognition model with a network structure of 1020-42-17-7-2. Its overall recognition accuracy reached more than 90%, and through applying the principal component analysis method to parameters by dimensionality reduction, the pig cough sound recognition accuracy of the DBN model was improved by 1.68%, indicating the great potential of deep learning algorithms in the study of pig cough sound recognition.

After the development of models such as AlexNets, VGG and ResNets [24–26], Convolutional Neural Networks (CNN) adopt a more complex network structure and deeper layers to pursue network models with higher performance, but at the same time, it also aggravates problems such as the vanishing-gradient problem. One of the most compelling advantages of DenseNets is that it alleviates the vanishing-gradients problem [27]. Although deep learning algorithms have been applied to pig cough recognition in previous studies, the improvement of the performance of deep learning algorithms has always been a vacancy in pig cough recognition. Therefore, in this study, we intend to construct a pig cough sound recognition model using the DenseNets algorithm and improve the accuracy of the model for pig cough sound recognition by incorporating an attention mechanism.

In Section 2, we build a pig sound signal acquisition system and use it to collect the sound data required for the test. Section 3 describes the process of sound sample preprocessing and extracting to Mel Frequency Cepstral Coefficient (MFCC) parameters of different dimensions. Section 4 introduces the improvement of Dense Block through SENets' attention module and further obtains the SE-DenseNet pig cough sound identification model required by the experiment. Section 5 presents the experimental results and discussion, which focuses on the analysis of the performance comparison of MFCC feature parameters in different dimensions and the analysis of the performance of the improved model. Section 6 is the conclusion of this study.

## 2. Materials and Methods

### 2.1. Pig Sound Signal Acquisition

In order to obtain high quality pig sound signals as the training data of the recognition model, this chapter designed a pig sound signal acquisition system to build a data base for the study of a pig cough sound recognition model.

### 2.1.1. Acquisition System Flow Design

The overall design flow of the pig sound signal acquisition system is shown in Figure 1. After the system is powered on, the system is first initialized and parameters are set, which include the pre-acquisition duration or the working target of the pre-acquisition memory; then, the working mode of the acquisition system will be selected between the normal mode and the backup mode according to the network communication.

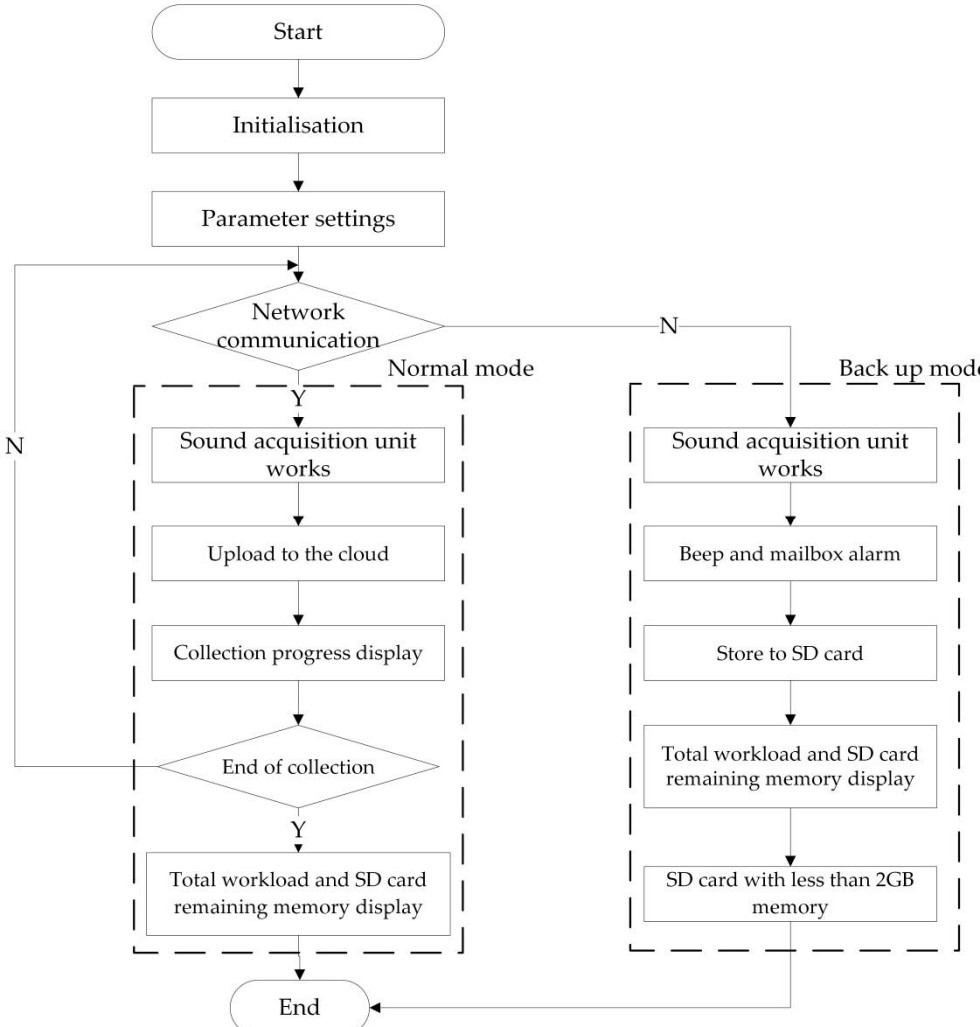

**Figure 1.** The flow of the pig sound signal acquisition system.

After the normal mode is enabled, the sound acquisition unit will acquire the sound signal, upload the collected audio signal data to the cloud for saving, and display the acquisition progress on the human-computer interaction interface until the preset acquisition target is reached and the acquisition is finished; after the backup mode is working, the audio acquisition unit will also acquire the sound signal, but the acquisition system will issue a beep and mailbox alarm to remind the staff to pay attention to the memory of the SD memory card (at this time, the saving mode of the acquisition system is local SD memory card saving) and stop the acquisition when the memory of the SD memory card is less than 2GB.

### 2.1.2. Introduction of Sound Acquisition Unit

The sound signal captured by the microphone array contains the spatial information of the sound source, has the advantages of spatial selectivity, eliminating many interferences in the environment and reducing the influence of echoes and improving the quality of the sound signal. In this study, the M260C Microphone Array with six SPA1687LR5H-1 microphone components was selected as the sound acquisition node of the pig sound signal acquisition system, which not only circumvents the shortcomings of traditional recorders in capturing sound signals, such as insufficient memory and high labor consumption, but also improves the quality of the sounds. Table 1 shows the hardware parameters of the M260C Microphone Array.

**Table 1.** Hardware parameters of the M260C Microphone Array.

| Product Name | M260C Microphone Array |
|---|---|
| PCB size | 79.5 × 1.2 mm |
| Sensitivity | −38 dBV/Pa |
| Signal-to-noise ratio | 65 dB |
| Operating voltage | 3.3 V |
| Operating current | 0.8 mA |
| Microphone model | SPA1687LR5H-1 |

2.1.3. Core Processor Unit

Since the data collected by the sound acquisition node need to be uploaded to the cloud for storage through the core processing unit, which requires a lot of data computation, the Raspberry Pi 4B board is used as the core processing unit in this study. Because the Raspberry Pi 4B development board is small, low cost and highly developable at a later stage, and it is an embedded device with microcomputer control, which is equipped with control, storage and communication functions, it meets the core processing unit requirements of this study. The hardware parameters of the Raspberry Pi 4B are shown in Table 2.

**Table 2.** Hardware parameters of the Raspberry Pi 4B board.

| CPU | 1.5GHz 4-Core BroadcomBCM2711BO(Cortex A-72) |
|---|---|
| GPU | 500 MHz VideoCore VI |
| USB port | USB2.0 × 2 + USB3.0 × 2 |
| Human-computer interaction port | micro HDMI ports × 2 lane MIPI DSI display port × 2 |
| Operating voltage | 5 V |
| Operating current | 3 A |
| SD memory card | 4 GB LPDDR4 |

*2.2. Data*

The collection site was Yabuli Pig Breeding Center, Harbin City, Heilongjiang Province. A total of 705 pig sound signals were used in this study, all from six pigs with frequent coughs caused by respiratory diseases. The pig sound acquisition system was fixed at about 1 m directly in front of the pens, with the sampling rate set at 16 kHz, a sampling accuracy of 16 bit, and the duration of each pig sound signal less than 1 h. The collected pig sound signals were classified and labeled. Pig sound signals include cough, grunts, squeal and snort, etc. However, the essence of the pig cough identification problem is a binary classification problem, so when classifying and labelling the data, only two categories of pig coughs and non-coughs were classified. The grunt, squeal and snort pig acoustic signal was marked as not cough, and 451 pig coughs and 994 non-coughs were finally obtained.

**3. Data Processing**

*3.1. Speech Enhancement Based on Spectral Subtraction*

The original pig sound signals collected in the piggery are superimposed with a large amount of background noise, such as footsteps and metal impact sound, which affect the model recognition effect. Therefore, in order to improve the model recognition performance, it is necessary to enhance the speech of the original pig sound signals collected, so as to improve the sound quality. Figure 2 shows the frequency analysis of environmental noise and pig cough sound signals in pig farms.

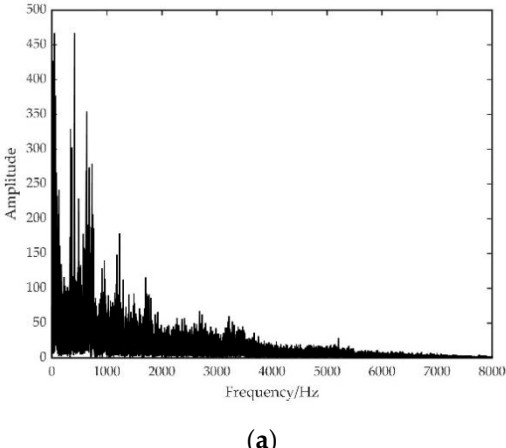

(**a**)

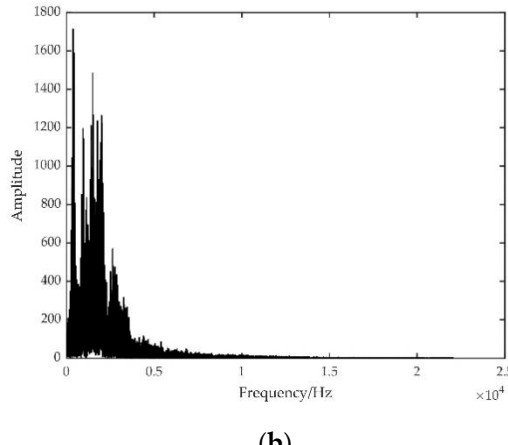

(**b**)

**Figure 2.** Frequency analysis of environmental noise and pig cough sound signals in pig farms: (**a**) the spectrum of the environmental noise of the pig farm; (**b**) the spectrum of the pig cough sound signals.

It can be seen from the Figure 2 that the pig cough frequency is in the range 0.3–8000 Hz, while the environmental noise is mainly concentrated below 6000 Hz. It is difficult for the digital filter to effectively denoise the additive noise. In this paper, the spectral subtraction [28] with good additive noise processing effect was selected for spectral subtraction of pig sound signals. The mathematical model of a pig sound signal with noise is shown in Equation (1):

$$y(t) = s(t) + d(t) \tag{1}$$

where $y(t)$ represents the pig sound signal with noise, $s(t)$ represents pure pig sound signal, and $d(t)$ represents noise.

Fast Fourier Transform (FFT) is performed on $y(t)$, $s(t)$ and $d(t)$ to obtain frequency-domain expressions of pig acoustic signals with noise in $Y(w)$, $S(w)$ and $D(w)$, as shown in Equation (2):

$$Y(w) = S(w) + D(w) \tag{2}$$

where $(w)$ represents the window operation. In this paper, Hanning window is selected as the window function, and the window length is 256. The Hanning window function is shown in Equation (3):

$$w(n) = \frac{1}{2} - \frac{1}{2}\cos\left(\frac{2\pi}{N}n\right) \tag{3}$$

where $N$ represents the serial number of the sampling point, $n$ is the total number of signal sampling points, and $n = 0, \ldots, n-1$.

The additive noise in the piggery is non-stationary because the pig sound signal and the additive noise are unrelated, so Equation (4) is obtained.

$$|S(\omega)|^2 = |Y(\omega)|^2 - |D(w)|^2 \tag{4}$$

where $|S(\omega)|^2$ represents the power spectrum of the pure pig sound signal, $|Y(\omega)|^2$ represents the power spectrum of the pig sound signal with noise, and $|D(w)|^2$ represents the power spectrum of noise. Based on the feature that human hearing is insensitive to sound phase changes, the pig sound signal $S(\omega)$ is subjected to discrete Inverse Fast Fourier Transform (IFFT) to obtain the pure pig sound signals after speech enhancement via spectral subtraction, as shown in Equation (5):

$$\hat{S}(\omega) = |S(\omega)|e^{j\theta_y(w)} \tag{5}$$

where $\hat{S}(\omega)$ represents the estimated value of the pure pig sound signal, and $\theta_y(w)$ represents the phase of noisy pig sound $Y(\omega)$ signal.

Figure 3 shows the comparison of pig cough sound signals and three non-pig cough sound signals before and after speech enhancement based on spectral subtraction. The left side shows the waveforms before speech enhancement, and the right side shows the waveforms after speech enhancement. As can be seen from the figure, spectral subtraction has an obvious effect on pig sound signal speech enhancement and greatly retains effective information, improves audio quality and lays a foundation for the construction of recognition model.

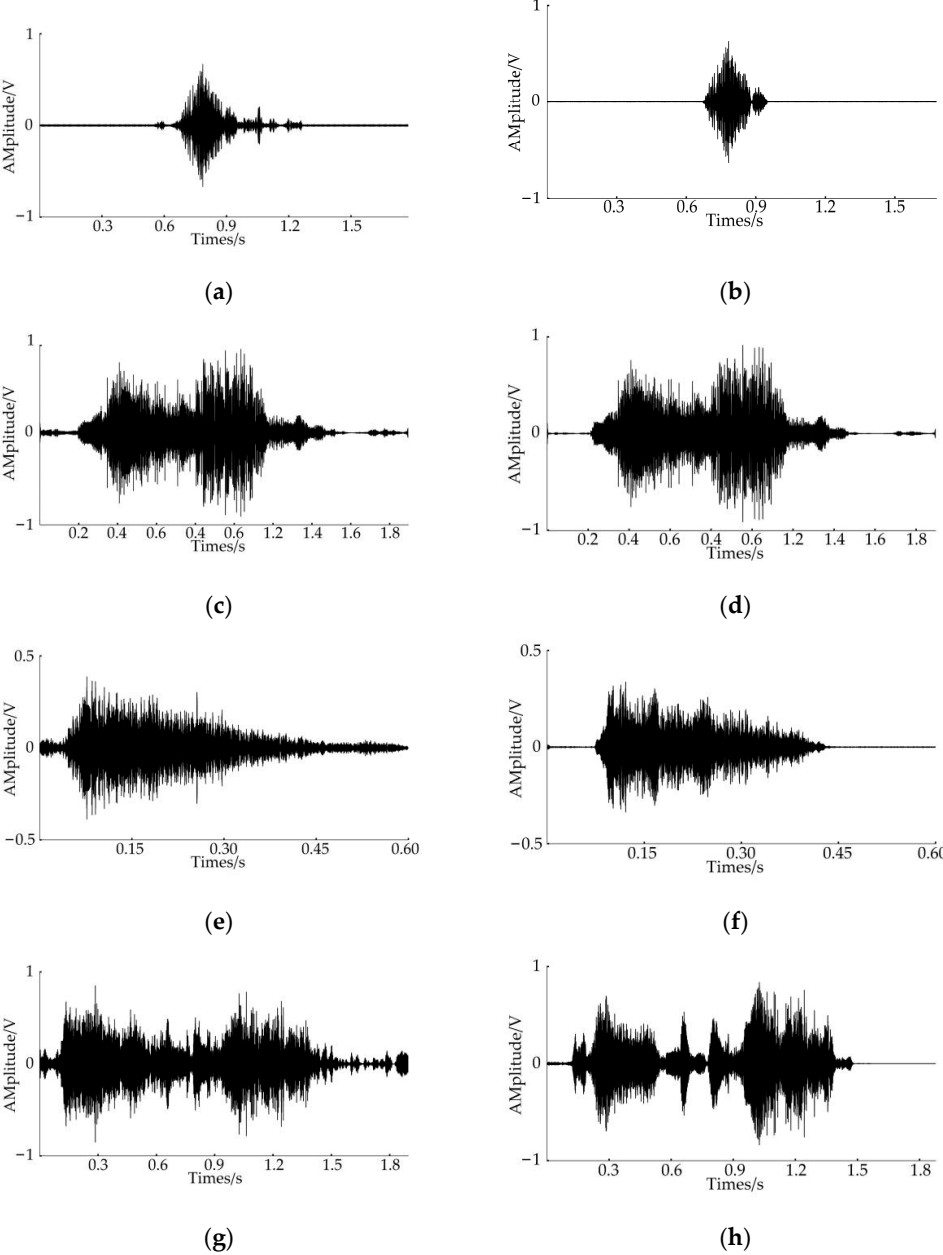

**Figure 3.** The comparison before and after speech enhancement based on spectral subtraction: (**a**) pig cough sound waveform before speech enhancement; (**b**) pig cough sound waveform after speech enhancement; (**c**) non-pig cough sound 1 waveform before speech enhancement; (**d**) non-pig cough sound 1 waveform after speech enhancement; (**e**) non-pig cough sound 2 waveform before speech enhancement; (**f**) non-pig cough sound 2 waveform after speech enhancement; (**g**) non-pig cough sound 3 waveform before speech enhancement; (**h**) non-pig cough sound 3 waveform after speech enhancement.

### 3.2. Endpoint Detection

There are some silent segments in speech enhanced pig sound signals, which will increase the amount of invalid data and interfere with the recognition effect. Therefore, in order to improve the recognition accuracy of the model, it is necessary to check endpoint detection of the pig sound signals to find out the starting and ending points of the valid information in the pig sound signals, eliminate the invalid information segments and maximize the retention of the valid information segments. In this paper, a single-parameter double threshold endpoint detection method based on short time energy is used to detect the pig sound signals. Formula 1 for calculating the short-term energy of frame *i* is shown in Equation (6).

$$E(i) = \sum_{n=0}^{L-1} x_i^2(n) \tag{6}$$

where *L* represents the frame length, *n* is the serial number of the sampling point, and *i* is the serial number of the frame. The waveform of the pig sound signal after endpoint detection is shown in Figure 4.

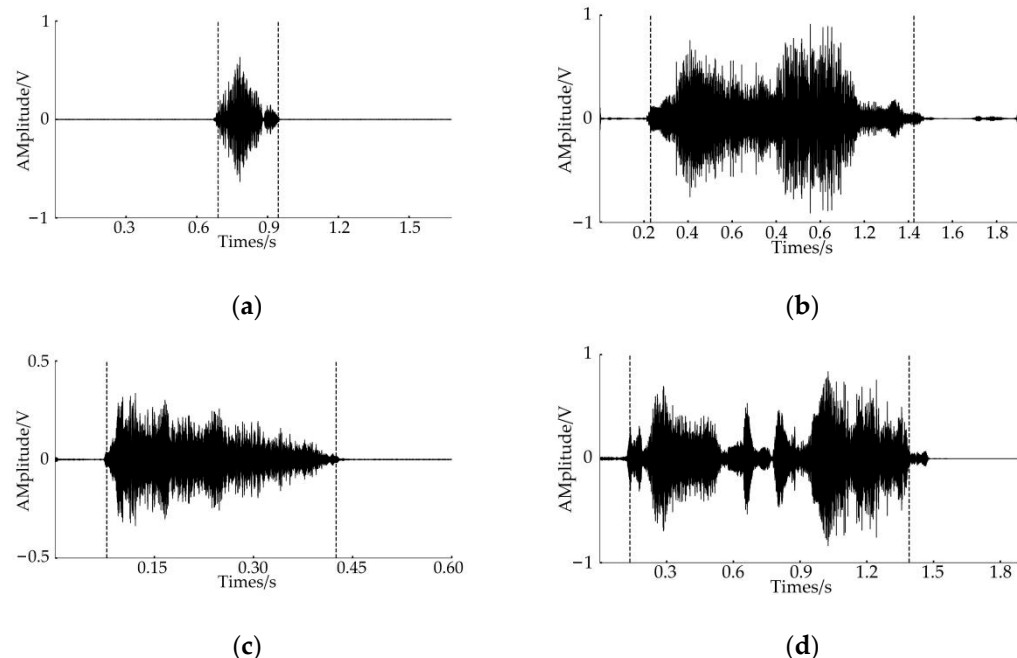

**Figure 4.** Endpoint detection results of pig sound signals: (**a**) pig cough sound endpoint detection results; (**b**) non-pig cough sound 1 endpoint detection results; (**c**) non-pig cough sound 2 endpoint detection results; (**d**) non-pig cough sound 3 endpoint detection results.

### 3.3. MFCC Parameters Extraction

A characteristic of the human ear's hearing of sound signals is that the volume of sound is not linearly proportional to the frequency. According to the mechanism of human ear hearing, the Mel Frequency Cepstral Coefficient (MFCC) [29] maps the linear spectrum of the sound signal to the nonlinear mel spectrum and analyzes the spectrum characteristics of the sound according to human ear hearing. The MFCC is one of the most commonly used parameters in the audio analysis field because of its strong anti-noise ability. The relationship between mel frequency and actual frequency [30] is as follows:

$$F_{\mathrm{Mel}} = 2595 \lg\left(1 + \frac{f}{700}\right) \tag{7}$$

where $F_{\mathrm{Mel}}$ represents the mel frequency, and *f* represents the actual frequency (unit: Hz).

MFCC parameters can reflect the static characteristics of pig sound signals. The ΔMFCC parameter is the first-order difference of the MFCC, which can describe the relationship between two adjacent frames of pig sound signals. The $\Delta^2$MFCC parameter is the second-order difference of the MFCC, which describes the relationship between three adjacent frames of pig sound signals. Both ΔMFCC and $\Delta^2$MFCC reflect the dynamic characteristics of pig sound signals. The 13-dimensional parameters of MFCC, ΔMFCC and $\Delta^2$MFCC are extracted, respectively, and they are spliced into a new set of parameters in any combination, which can fully reflect the characteristics of pig sound signals. The overall extraction process of MFCC parameters is shown in Figure 5.

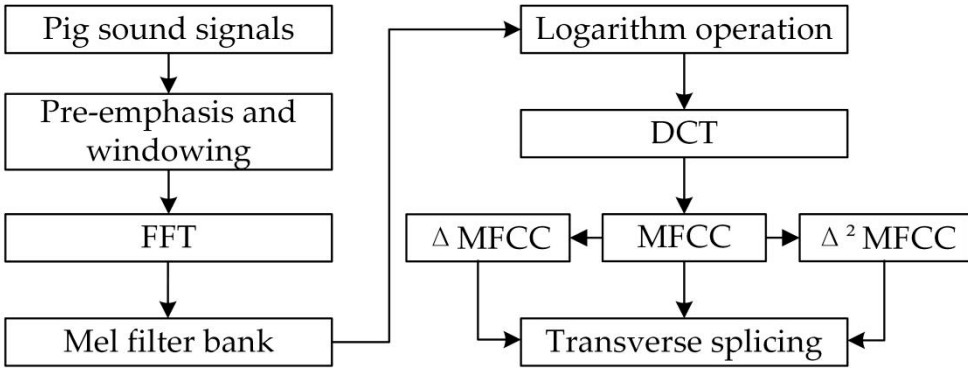

**Figure 5.** MFCC parameter extraction process.

The original pig sound signal can emphasize the high frequency formants in the signal and increase the output signal-to-noise ratio (SNR) by means of pre-emphasis without influence on the noise. The pre-emphasis formula is as follows:

$$y(n) = x(n) - ax(n-1) \tag{8}$$

where *y(n)* represents the pre-emphasis pig sound signal, *x(n)* represents the original pig sound signal, *n* represents the serial number of the sampling point, and *a* is the pre-emphasis coefficient, which is set as 0.9 in this paper.

Because the pig sound signals have the characteristic of short-term stability, the pig sound signals should be windowed before FFT, and some overlapping areas should be set between frames to avoid the signal leakage at the window boundary during FFT. In this paper, the window function is a Hamming window, with a window length of 25 ms and a window step of 10 ms. The mathematical model of the Hamming window [30] is as follows:

$$W(n) = 0.54 - 0.46 \cos\left(\frac{2\pi n}{N-1}\right) \tag{9}$$

where *n* is the serial number of sampling points, and *N* is the total number of sampling points. The pre-processed pig sound signals are transformed via FFT to obtain the frequency domain data. The formula is as follows:

$$X(i,k) = \text{FFT}[x_i(m)] \tag{10}$$

where $x_i$ is the pre-processed pig sound signal, *I* is the frame sequence number, and *k* is the number of points of FFT. FFT is set to 512 in this paper.

The core of the way the MFCC mimics human ear hearing is that a bandpass filter with triangular filtering characteristic is set in the mel filter bank. In the MEL frequency, these filters are equal bandwidth, and the frequency response of the bandpass filter is as follows:

$$H_m(k) = \begin{cases} 0 & k < f(m-1) \\ \frac{2[k-f(m-1)]}{[f(m)-f(m-1)][f(m+1)-f(m-1)]} & f(m-1) \le k \le f(m) \\ \frac{2[f(m+1)-k]}{[f(m+1)-f(m)][f(m+1)-f(m-1)]} & f(m) \le k \le f(m+1) \\ 0 & k > f(m+1) \end{cases} \tag{11}$$

where $H_m(k)$ represents the bandpass filter, $k$ represents the filter serial number, $f(m)$ represents the center frequency of the mel filter, $0 \le m \le M_0$, $M_0$ is the number of bandpass filters, and the number of bandpass filters is set as 26 in this paper. The expression for $f(m)$ is as follows:

$$f(m) = \frac{N}{F_s} F_{\text{Mel}}^{-1} \left[ F_{\text{Mel}}(f_1) + m \frac{F_{\text{Mel}}(f_h) - F_{\text{Mel}}(f_1)}{M_0 + 1} \right] \tag{12}$$

where $F_{\text{Mel}}^{-1}$ is the inverse of $F_{\text{Mel}}$, $N$ is the window length of the Fourier transform, set to 25 ms, $F_s$ is the sampling frequency, $f_1$ is the lowest frequency of the filter, and $f_h$ is the highest frequency of the filter. Set $f_1$ to 0 Hz and $f_h$ to 8000 Hz, respectively. The formula for $F_{\text{Mel}}^{-1}(b)$ is as follows:

$$F_{\text{Mel}}^{-1}(b) = 700 \left( 10^{\frac{b}{2595}} - 1 \right) \tag{13}$$

The energy of the input signal in each mel filter is calculated as the power value of the discrete power spectrum $E(i,k)$ passing through mel filter $H_m(k)$. The formula is as follows:

$$S(i,m) = \sum_{k=0}^{N-1} E(i,k) H_m(k) \tag{14}$$

where $0 \le m \le M_0$, $S(i,m)$ represents the energy sum of the pig sound signal at frame $i$ in the $m$-th frequency band of the mel domain, $E(i,k)$ is the discrete power spectrum of the pig sound signals, and the calculation formula of $E(i,k)$ is as follows:

$$E(i,k) = [X_i(k)]^2 \tag{15}$$

After taking the logarithm of the energy of $E(i,k)$, the discrete cosine transform (DCT) is done to obtain the MFCC parameters, and the calculation formula is as follows:

$$C(j) = \sqrt{\frac{2}{M_0}} \sum_{m=1}^{M_0} \lg[S(i,m)] \cos\left( j \frac{\pi(2m-1)}{M_0} \right) \tag{16}$$

where $1 \le j \le Y$, $Y$ represents the parameter dimensions of the MFCC output. In this paper, $Y = 13$, and 13-dimensional MFCC parameters are output. Then, the first and second difference parameters based on the pig sound signals MFCC are obtained by $C(j)$, and the calculation formulas are as follows:

$$dC(j) = \frac{C_{i-1}(j) - C_{i+1}(j)}{2} \tag{17}$$

$$DC(j) = \frac{C_{i-2}(j) - C_{i+2}(j)}{2} \tag{18}$$

where $dC(j)$ represents the parameters ΔMFCC of MFCC first-order difference, and $DC(j)$ represents the parameters $\Delta^2$MFCC of MFCC second-order difference.

Figure 6 shows the three-dimensional view of the parameters of pig cough sounds and non-pig cough sounds. The left column respectively show MFCC 3-dimensional diagrams, the middle column respectively show ΔMFCC 3-dimensional diagrams, and the right column respectively show $\Delta^2$MFCC 3-dimensional diagrams. Each row from top to bottom

are three-dimensional maps of pig cough sound, non-pig cough sound 1, non-pig cough sound 2 and non-pig cough sound 3.

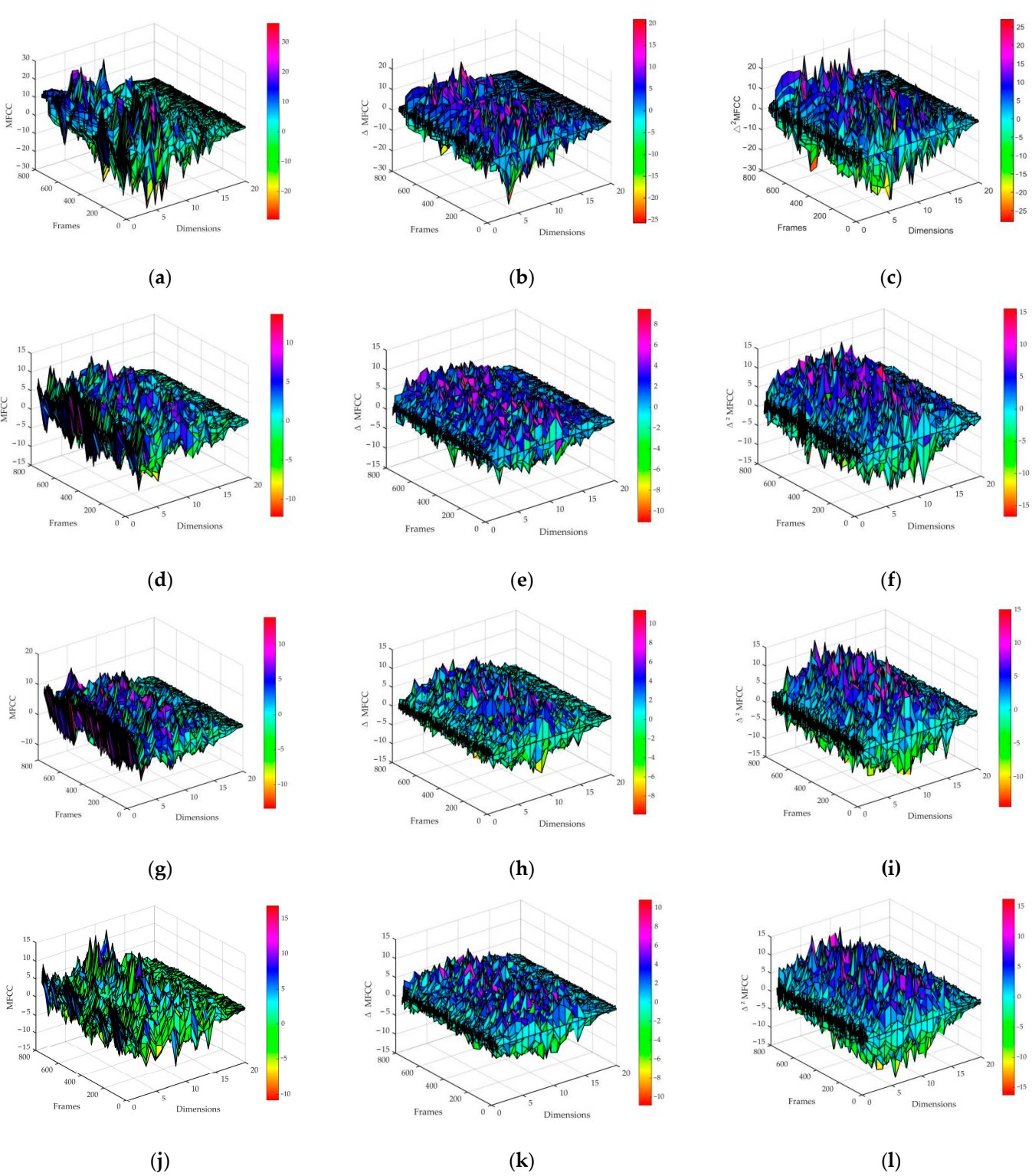

**Figure 6.** MFCC 3D diagram of pig sound signal. (**a**) Pig cough sound MFCC; (**b**) pig cough sound ΔMFCC; (**c**) pig cough sound $\Delta^2$MFCC; (**d**) non-pig cough sound 1 MFCC; (**e**) non-pig cough sound 1 ΔMFCC; (**f**) non-pig cough sound 1 $\Delta^2$MFCC; (**g**) non-pig cough sound 2 MFCC; (**h**) non-pig cough sound 2 ΔMFCC; (**i**) non-pig cough sound 2 $\Delta^2$MFCC; (**j**) non-pig cough sound 3 MFCC; (**k**) non-pig cough sound 3 ΔMFCC; (**l**) non-pig cough sound 3 $\Delta^2$MFCC.

The parameter values of pig cough sound signals and non-pig cough sound signals in different dimensions are significantly different, and the differences are most obvious before 13 dimensions. Therefore, MFCC, ΔMFCC and Δ²MFCC were transverse spliced to obtain 6 groups of parameters that could respectively characterize the static, dynamic and mixed characteristics of pig sound signals. They are 13-dimensional MFCC, 13-dimensional ΔMFCC, 13-dimensional Δ²MFCC, 26-dimensional MFCC + ΔMFCC, 26-dimensional MFCC + Δ²MFCC, 26-dimensional ΔMFCC + Δ²MFCC and 39-dimensional MFCC + ΔMFCC + Δ²MFCC.

## 4. Model Building

### 4.1. DenseNets Model

DenseNets transfers the features of all layers via dense connections, so that it can explore the network model with better performance and deeper layers under the premise of less parameters and computation. DenseNets is mainly composed of a dense block and transition layer. The DenseNets model structure is shown in Figure 7.

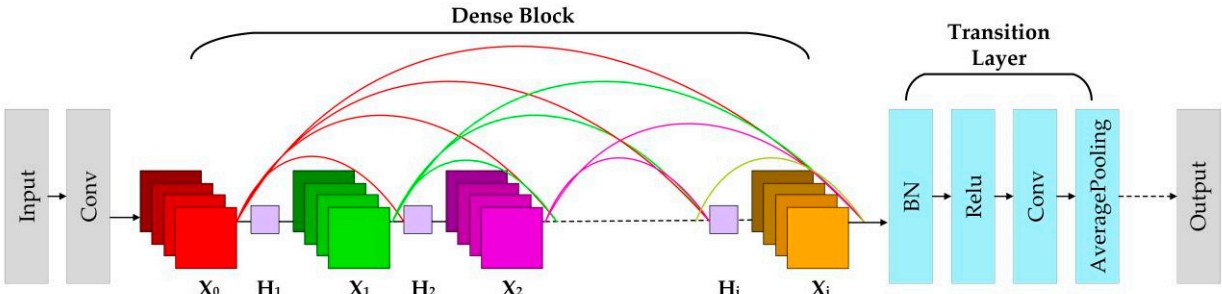

**Figure 7.** Structure diagram of DenseNets.

The DenseNets model is composed of multiple dense blocks and transition layer. $X_0$ is the input layer of a dense block, $X_i$ is the output feature of $H_i$, and the output feature of $X_i$ is shown in Equation (19).

$$X_i = H_i([X_0, X_1, \cdots, X_{i-1}]) \tag{19}$$

where $H_i$ represents the nonlinear transformation function at layer $i$ composed of Batch Normalization (BN), Rectified Linear Units (ReLU) and Convolution (Conv).

The transition layer consists of BN, ReLU, Conv($1 \times 1$) and average pooling. Its role is to integrate the output characteristics of the dense block in front and the huge amount of parameter reduction of data through Conv($1 \times 1$) in order to reduce the number of parameters and then reduce the amount of calculation, so that the model is more lightweight.

### 4.2. Squeeze and Excitation Networks

Traditional Convolutional Neural Networks (CNNs) acquire global features on the local receptive field, ignoring the detailed features between channels. Squeeze-and-Excitation Networks (SENets) [31] can achieve feature re-calibration, which can improve the effective feature extraction ability between different channels of the CNN, suppress the extraction of useless features and improve the recognition accuracy of the algorithm. Figure 8 shows the structure diagram of SENets.

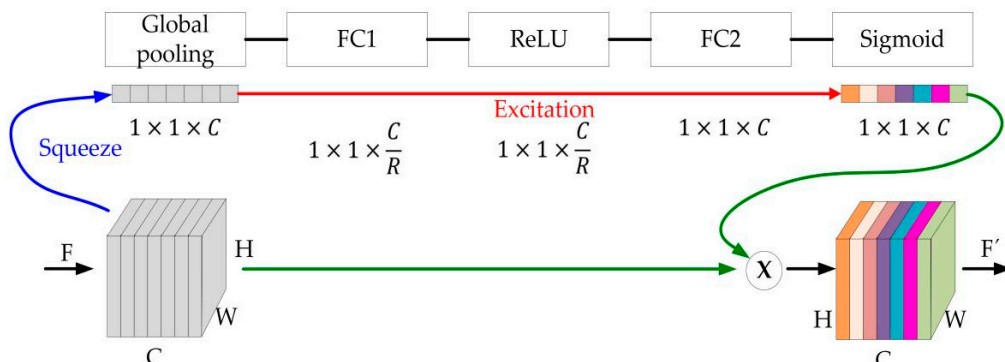

**Figure 8.** Structure of SENets attention module.

The SENets module is divided into two parts: squeeze and excitation. The input *F* is the output of the previous CNN, and the feature-maps size is $H \times W \times C$. The original feature map is squeezed into $1 \times 1 \times C$ features via global pooling, which is the squeeze part of SENets, and the formula is as follows:

$$S_1 = \frac{1}{W \times H} \sum_{j=1}^{W} \sum_{k=1}^{H} u_i(j,k) \tag{20}$$

where *H*, *W* and *C* represent the height, width and number of channels of the feature-maps, respectively, and $u_i(j,k)$ represents the element $(j,k)$ at the *i*-th channel position, $i \in C$.

The excitation part is composed of FC1, ReLU, FC2 and Sigmoid. FC1 and FC2 are the fully connected layers. ReLU and Sigmoid are the activation functions, and the formulas are as follows:

$$\text{ReLU} = \begin{cases} x & x \geq 0 \\ 0 & x < 0 \end{cases} \tag{21}$$

$$\text{Sigmoid} = \frac{1}{1 + e^{-x}} \tag{22}$$

Feature maps are compressed into $\frac{C}{R}$ by FC1. This paper takes *R* = 4, after Relu and FC2, the size of the feature maps is reduced to *C*, and then the value between *C* 0 and 1 is obtained by sigmoid, which is equivalent to the weight of *C* feature channels, and the weight is multiplied by the input feature map *F* to obtain the output feature map *F'*. The calculation process is as follows:

$$S_2 = \text{ReLU}(W_1 S_1) \tag{23}$$

$$S_3 = \text{Sigmoid}(W_2 S_2) \tag{24}$$

$$F' = S_3 F \tag{25}$$

where *F* represents the input feature-maps, $W_1$ represents the parameters of FC1, $W_2$ represents the parameters of FC2, and $W_3$ represents the output of the SENets attention module.

*4.3. Improved Dense Block Based on SENets Attention Module*

The core of DenseNets is to repeatedly use the dense block module in the network structure, extract features through Conv(1 × 1) and Conv(3 × 3) kernels and strengthen the propagation efficiency of features in the form of dense links in the module, so that features can be reused throughout the network transmission process. Therefore, this paper improves the dense block module. Each dense block is spliced by Conv(1 × 1) and Conv(3 × 3), and after the SENets attention module is embedded to Conv(3 × 3), the output of the original dense block is used as the input of SENets. The dense block re-calibrates features reused by using weights to improve the utilization rate of effective features by DenseNets and ignoring irrelevant features to improve the recognition accuracy of the algorithm. The structure of the SE-dense block is shown in Figure 9.

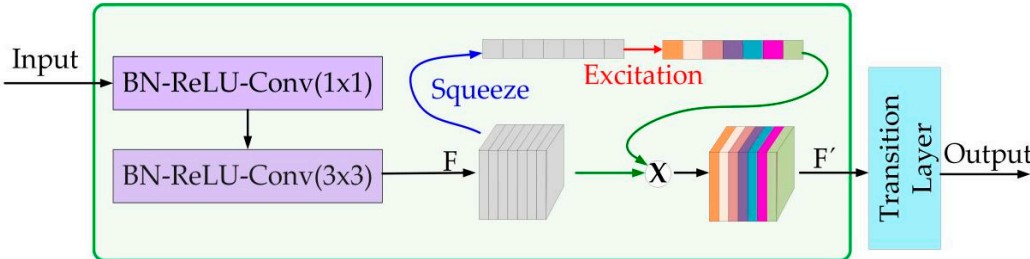

**Figure 9.** SE-dense block structure.

### 4.4. SE-DenseNet-121 Pig Cough Sound Recognition Model

The MFCC parameters of pig sound signals are taken as model inputs, and the input feature parameters are initially extracted through the convolutional layer of the Conv (7 × 7) kernel, and then the improved four sets of SE-dense blocks and three transition layers are alternately stitched to transmit the valid features of reuse to the classifier and finally get the classification results. The number of SE-dense blocks in the four groups is 6, 12, 24, and 16, respectively, and the SE-DenseNet-121 pig cough sound recognition model is shown in Figure 10.

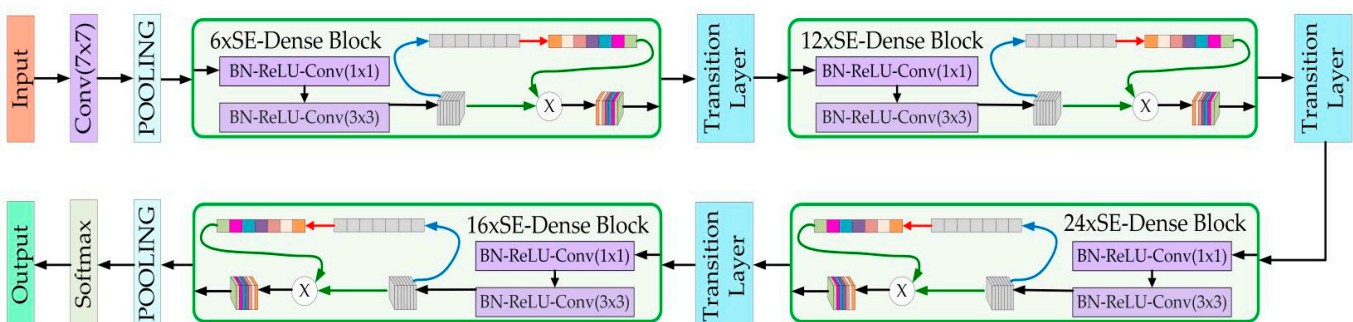

**Figure 10.** SE-DenseNet-121 pig cough sound recognition model.

## 5. Experiment

### 5.1. Experimental Background

5.1.1. Experimental Environment

The experiment uses an Intel(R) Core(TM) i7-10870H CPU @ 2.20GHz 2.21 GHz hardware processor, an NVIDIA GeForce RTX3060 Laptop GPU, cuDNN8.1 as a deep neural network acceleration library, and the deep learning Tensorflow2.5 framework is implemented using Python3.7 language.

5.1.2. Model Parameters Setting

The Stochastic Gradient Decent (SGD) optimization function is selected for CNN model training, and the categorical_crossentropy function is selected for the loss function, which is set to 64 batch_size and the number of iteration rounds is 50. SGD expressions are as Equation (26), and categorical_crossentropy function expressions are as Equation (27):

$$w_{t+1} = w_t - \eta \nabla_w J(w_t) \tag{26}$$

where $\nabla_w J(w_t)$ is the gradient value of $J(w_t)$ on $w_t$, and $\eta$ is the learning rate.

$$\text{loss} = -\frac{1}{n} \sum_{i=1}^{n} \sum_{k=1}^{m} \hat{y}_{ik} \boldsymbol{log} y_{ik} \tag{27}$$

where $n$ represents the total number of samples, $m$ is the number of model classifications, $y_{ik}$ is the probability that the $i$-th sample of the model is predicted to be $k$, and $\hat{y}_{ik}$ is the value of the $i$-th sample label in the $k$-th classification.

5.1.3. The Evaluation Index

In order to ensure the multi-angle performance analysis of the recognition model, according to the classification results of each pig sound signal and the actual classification labels of the model, the binary classification problem can obtain four classification combination results, which are classified as *TP*, *FP*, *FN* and *TN*, and the confusion matrix is used to analyze the classification of the model as shown in Table 3.

**Table 3.** Confusion matrix.

|  | **True** | **False** |
|---|---|---|
| **Positive** | TP | FP |
| **Negative** | TN | FN |

*Accuarcy, loss, precision, recall* and *F1 score* are used as the evaluation indicators of the model, and the calculation formula of each evaluation index is as follows:

$$Accuracy = \frac{TP + TN}{TP + TN + FP + FN} \times 100\% \tag{28}$$

$$Recall = \frac{TP}{TP + FN} \times 100\% \tag{29}$$

$$Precision = \frac{TP}{TP + FP} \times 100\% \tag{30}$$

where *TP* is the correct number of pig cough sound classifications, *TN* is the number of non-cough sounds correctly classified, *FP* is the number of misclassified pig cough sounds, and *FN* is the number of misclassified non-cough sounds.

*F1 score* is an important evaluation index in the performance analysis of the classification model, which can intuitively reflect the performance of the models in all directions, and the calculation formula is as follows:

$$F1_k = 2 \times \frac{P_k \times R_k}{P_k + R_k} \tag{31}$$

$$F1 = \frac{1}{n} \sum_{k=1}^{n} F1_k \tag{32}$$

where *n* is the number of categories, and $F1_k$ is the *F1 score* of the *k*-th category.

*5.2. Feature Parameter Performance Comparison*

By training the model with different feature parameter combinations as input features, the pig cough sound recognition results based on the DenseNet-121 model are shown in Table 4.

The performance of the parameters was analyzed and compared according to the four evaluation indexes of recognition accuracy, recall, precision and *F1 score*. The recognition accuracy of MFCC parameters in the pig cough sound recognition task was 92.4%, which was much higher than the 80.7% and 85% of ΔMFCC and Δ²MFCC, and the results of other evaluation indicators are also higher than ΔMFCC and Δ²MFCC parameters, with a recall of 98.4%, an accuracy of 96.4% and an *F1 score* of 97.4%, indicating that compared with dynamic characteristics, the differences between the static characteristics of pig sound signals are greater, and the recognition model can more easily distinguish between each pig acoustic signal through the difference in static characteristics. Finally, the correct classification results are obtained. The recognition results of the 26 dimensional ΔMFCC + Δ²MFCC combined parameters can also confirm this conclusion.

**Table 4.** Recognition results of DenseNet-121 for feature parameters of different dimensions.

| Feature | Epoch | Accuracy | Recall | Precision | *F1 Score* |
|---|---|---|---|---|---|
| MFCC | 50 | 92.1% | 98.4% | 96.4% | 97.4% |
| ΔMFCC | 50 | 80.7% | 95.9% | 94.8% | 95.3% |
| $\Delta^2$MFCC | 50 | 85% | 97.2% | 95.2% | 96.2% |
| MFCC + ΔMFCC | 50 | 92.9% | 98.6% | 96.6% | 97.6% |
| MFCC + $\Delta^2$MFCC | 50 | 91.5% | 98.5% | 96.7% | 97.6% |
| ΔMFCC + $\Delta^2$MFCC | 50 | 82.7% | 96.9% | 94.9% | 95.9% |
| MFCC + ΔMFCC + $\Delta^2$MFCC | 50 | 91.7% | 98.4% | 96.9% | 97.7% |

For the 13-dimensional parameters ΔMFCC and $\Delta^2$MFCC that can reflect the dynamic characteristics of pig sound signals, the recognition results of $\Delta^2$MFCC are better than ΔMFCC. Among the four evaluation indicators, the recognition accuracy of $\Delta^2$MFCC is 85%, which is higher than that of ΔMFCC with 80.7% recognition accuracy. Therefore, it can be concluded that compared with ΔMFCC, the recognition model can extract more effective features in $\Delta^2$MFCC for the recognition of a pig coughing sound. However, the recognition results of the combined parameters of ΔMFCC + $\Delta^2$MFCC based on the four evaluation indexes are all higher than that of the 13-dimensional ΔMFCC and lower than that of the $\Delta^2$MFCC, indicating that the dimension of the parameters cannot directly affect the recognition results in a positive way, because the number of parameters of the 26-dimensional combined parameters is twice that of the 13-dimensional parameters, which is equivalent to that when the model extracts features from the $\Delta^2$MFCC. On the basis of the original parameters, the $\Delta^2$MFCC feature parameters with the same number of parameters as ΔMFCC but fewer effective features are added. The number of parameters is doubled, but the proportion of effective features is reduced, which leads to the omission of some effective features in ΔMFCC during feature extraction of the recognition model.

For the combined feature parameters that can reflect both static and dynamic characteristics, the recognition accuracy of MFCC + ΔMFCC is 92.9%, and the recall rate is 98.6%, which is 1.4% and 0.1% higher than that of MFCC + $\Delta^2$MFCC, respectively. The accuracy of MFCC + ΔMFCC is 96.6%, which is 0.1% lower than that of MFCC + $\Delta^2$MFCC. The *F1 score* is 97.6%, indicating that when combined with MFCC parameters, $\Delta^2$MFCC can classify pig cough sounds more accurately, while the ability to classify non-cough sounds is not as good as ΔMFCC. The recognition accuracy and recall rate of MFCC + ΔMFCC are 1.2% and 0.2% higher than those of MFCC + ΔMFCC + $\Delta^2$MFCC, and the accuracy and *F1 score* are 0.3% and 0.1% lower than those of MFCC + ΔMFCC + $\Delta^2$MFCC, respectively. This indicates that when $\Delta^2$MFCC is splicing with MFCC + ΔMFCC to form a new 39-dimensional combined parameter MFCC + ΔMFCC + $\Delta^2$MFCC, the dynamic differences of pig sound signals in $\Delta^2$MFCC can make the prediction results of pig cough sound samples more reliable, but due to the excessive invalid features in $\Delta^2$MFCC, MFCC + ΔMFCC + $\Delta^2$MFCC misjudges a part of the cough sound signals as non-pig cough sound signals, resulting in a lower recognition accuracy and *F1 score*.

According to the research object of this paper and Equations (29) and (30), the recall is equivalent to the probability that the recognition model can correctly predict a pig cough sound among all pig cough sound samples, while non-pig cough sound samples have no effect on recall. The accuracy is the probability that the sample in which the recognition model predicts the pig cough sound is actually a pig cough sound, and the non-pig cough sound sample has an effect on the accuracy. The research of the pig cough sound recognition model is to realize the early warning of respiratory diseases of pigs in intensive housing pigs, detect and deal with the diseased pigs in time, improve animal welfare breeding and improve the productivity of pig farms. In the two evaluation indexes of recall and precision,

recall is not affected by non-pig cough sound samples, which can better reflect the practical application significance of parameters. Therefore, this paper believes that the combined parameters of MFCC + ΔMFCC has the best performance.

### 5.3. Model Performance Comparison

To evaluate the performance of the SE-DenseNet-121 pig cough sound recognition model, in the same dataset and experimental environment, SE-DenseNet-121, DenseNet-121, ResNet-34, VGG-16 and InceptionV1 take MFCC + ΔMFCC parameters as the input of the model to train the model. Figure 11 shows the accuracy variation curves of each model on the validation set with MFCC + ΔMFCC characteristic parameters as input after 50 epochs of iteration.

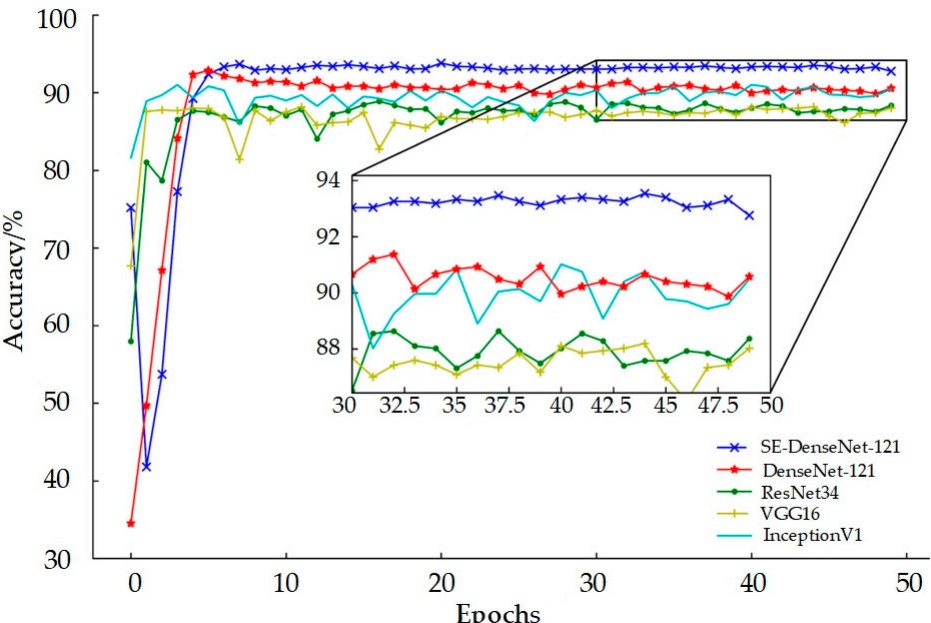

**Figure 11.** Accuracy variation curves of different models on the validation set with MFCC + ΔMFCC as parameters.

As can be seen in Figure 11, the convergence speed of SE-DenseNet-121 is slower than that of DenseNet-121. The recognition accuracy of SE-DenseNet-121 reaches 92.4% in the 6th iteration, while that of DenseNet-121 reaches 92.3% in the 5th iteration. After the above corresponding iteration epochs, the accuracy of the variation curves enters the shock period, and the accuracy is no longer greatly improved. This is because SE-dense block has more Conv(1 × 1) and Conv(3 × 3) than dense block, which increases the model parameter quantity and the calculation quantity, resulting in a slower convergence speed. After the 35th generation, the curves of ResNet34 and VGG16 nearly coincide, and the accuracy of convergence oscillates around 88%. After the 44th generation, the curves of DenseNet-121 and InceptionV1 nearly coincide, and the accuracy oscillates around 91% after convergence. However, DenseNet-121, ResNet-34, VGG-16 and InceptionV1 have large oscillations, and the oscillations have not improved significantly after about 40 epochs of iterations. Compared with DenseNet-121, ResNet-34, VGG-16 and InceptionV1, SE-DenseNet-121 has the smallest oscillation amplitude in the oscillation period, which is around 93.5% after 7 generations, and the recognition accuracy is stable at more than 93%. It shows that the SE-DenseNet-121 has the best performance in recognition accuracy and efficiency. Figure 12 shows the comparison of training and validation accuracy curve changes.

**Table 5.** Performance comparison of different models.

| Model | Epoch | Accuracy | Recall | Precision | *F1 Score* |
|---|---|---|---|---|---|
| SE-DenseNet-121 | 50 | 93.8% | 98.6% | 97% | 97.8% |
| DenseNet-121 | 50 | 92.9% | 98.6% | 96.6% | 97.6% |
| ResNet34 | 50 | 88.9% | 98.2% | 96% | 97.1% |
| VGG16 | 50 | 88.2% | 98.3% | 96.7% | 97.5% |
| InceptionV1 | 50 | 91% | 98.6% | 96.1% | 97.4% |

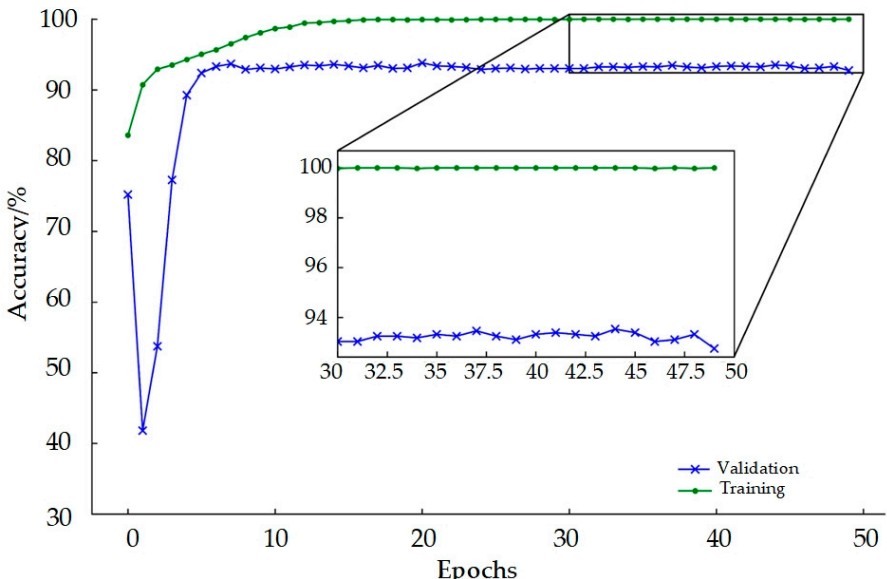

**Figure 12.** Comparison of training and validation accuracy curve changes. The MFCC + ΔMFCC parameters were taken as the model input, and the recognition accuracy, recall, precision and *F1 score* were taken as the evaluation indexes on different recognition models. The performance of the model was tested using 50 epochs of training iterations. The experimental results of pig cough sound recognition via different models are shown in Table 5.

Under the condition of MFCC + ΔMFCC as input parameters, the accuracy, precision and *F1 score* of the SE-DenseNet-121 model are increased by 0.9%, 0.4% and 0.2% compared with the DenseNet-121 model after 50 epochs of training, indicating that SE-DenseNet-121 can better extract effective features from pig sound signals through the SENet attention module and distinguish pig cough sounds from non-pig cough sounds. The model improvement is successful. The recognition accuracy, precision and *F1 score* of the SE-DenseNet-121 model were 93.8%, 97% and 97.8%, respectively, which were 4.9%, 5.6% and 2.8% higher than those of ResNet-34, VGG-16 and InceptionV1, respectively, and the recall was not less than that of the other four models, which was 98.6%. It is proved that the SENets attention module can improve the performance of pig cough sound recognition models, and the SE-DenseNet-121 model is the best pig cough sound recognition model.

## 5.4. Discussion

In terms of parameter extraction, we adopt a transversal splicing method different from that in the literature [23]. We combine MFCC, ΔMFCC and $\Delta^2$MFCC to form new parameters to pursue the best set of parameters. The best set of parameters is MFCC + ΔMFCC as verified by experiments.

In the task of pig cough recognition, the performance of the deep learning algorithm is better than that of the identification method in the literature [13–15]. Then, in order to seek higher recognition accuracy, we improved DenseNet-121. We added the SENets attention module to the DenseNet-121 model to improve the ability of effective feature extraction

between different channels. The accuracy of the SE-DenseNet-121 pig cough recognition model was 93.8%, and the performance improved significantly.

During the experiment, we found that due to the large dimensions of MFCC + ΔMFCC, MFCC + $\Delta^2$MFCC, ΔMFCC + $\Delta^2$MFCC and MFCC + ΔMFCC + $\Delta^2$MFCC, the number of feature parameters is too large, which increases the calculation amount of the model, and the performance of model is not efficient enough. Therefore, in the subsequent research, we will focus on the problem of MFCC parameters dimensionality reduction.

## 6. Conclusions

In this paper, the pig cough sound recognition model is improved via the SENets attention mechanism. We analyzed the characteristics of different dimensions of MFCC parameters in pig sound signals, verified the performance of parameters using the DenseNet-121 model and tested the SE-DenseNet-121 model by taking the optimal parameters as the input of the model. The results show that in the pig cough sound recognition task, the optimal parameter combination is MFCC + ΔMFCC with 26 dimensions. The accuracy, recall, precision and *F*1 *score* of the SE-DenseNet-121 model are 93.8%, 98.6%, 97% and 97.8%, respectively, and the accuracy, precision and *F*1 *score* of the SE-DenseNet-121 model are 0.9%, 0.4% and 0.2% higher than the DenseNet-121 model, respectively. This study provides reference significance for intelligent early warning of pig diseases and is expected to improve animal welfare and improve the productivity of pig farms.

**Author Contributions:** Conceptualization, H.S. (Hang Song) and B.Z.; methodology, H.S. (Hang Song) and B.Z.; software, H.S. (Hang Song); validation, H.S. (Hang Song), B.Z. and Z.Z.; formal analysis, H.S. (Hang Song); investigation, H.S. (Haonan Sun); resources, H.S. (Haonan Sun); data curation, H.S. (Haonan Sun); writing—original draft preparation, H.S. (Hang Song); writing—review and editing, H.S. (Hang Song); visualization, H.S. (Hang Song) and J.H.; supervision, H.S. (Hang Song) and J.H.; project administration, Z.Z.; funding acquisition, Z.Z. All authors have read and agreed to the published version of the manuscript.

**Funding:** This research was funded by the National Natural Science Foundation of China (NSFC), grant number 32201655.

**Data Availability Statement:** Not applicable.

**Acknowledgments:** The author thank Song Wang and Lan Wang for their assistance.

**Conflicts of Interest:** The authors declare no conflict of interest.

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
