# Peer review of "Research on Improved DenseNets Pig Cough Sound Recognition Model Based on SENets"

_electronics, doi:10.3390/electronics11213562_

Round 1
Reviewer 1 Report
The authors propose research on an improved DenseNets pig cough sound recognition model based on SENets for early detection of swine respiratory diseases. In this work, the DenseNet-212 recognition model was improved by using the SENets attention module to enhance the model's capability to extract the effective features from the pig sound signal. The manuscript is well written and the results are demonstrated properly. But there are some suggestions that need to be incorporated into the manuscript before consideration.
- In the abstract, only the final accuracy rate , precision, and F1 score are sufficient. No need to mention the parameters increased in percentage.
- Related literature works are properly discussed, but research gaps are missing in few works.
- Authors claim that dataset is recorded from the pigs but characteristics of the data set in not mentioned like frequency, length of the segment.
- The training vs validation graph of the best model needs to be incorporated into the manuscript.
- Confusion matrix of the best results needs to be presented.
- In the discussion section, the final obtained results must be compared to the existing works.
- The highlights and limitations of the work need to be included.
Reviewer 2 Report
The work is good, I recommend it for publication. But before that, I advise the authors to correct a few points.
1. General opinion - in the abstract and in the conclusion there is no description of the significance of the results obtained. "The rate of accuracy, precision and score were increased" - so what? What it allows/will allow?
2. line 24 : "With increased market demand, pork has become the most consumed meat in the world" - I advise you to add a reference to hard good work for such a statement in addition to [1].
3. line 72: MFCC - expand
4. In such work, in addition to the amplitude spectrum (Fig. 2), there must be a frequency analysis of the initial data before/after primary processing. I strongly recommend to add it with a description.
5. Eq. (7) - it is necessary to explain this formula, add a ref, for example, [arxiv 1003.4083]. Same goes for the Hamming window (eq.9).
6. Fig.5 - add color scale.
7. Other specialists have similar studies, for example, [20]. Why not compare the results obtained by general parameters characterizing the effectiveness of the application?
